# Diagnosis and Treatment of *Helicobacter pylori* Infection in Real Practice—New Role of Primary Care Services in Antibiotic Resistance Era

**DOI:** 10.3390/diagnostics13111918

**Published:** 2023-05-30

**Authors:** Enrique Alfaro, Carlos Sostres, Angel Lanas

**Affiliations:** 1Department of Gastroenterology, Lozano Blesa University Clinic Hospital, 50009 Zaragoza, Spain; 2Aragon Health Research Institute (IIS Aragón), 50009 Zaragoza, Spain; 3Biomedical Research Networking Center in Hepatic and Digestive Diseases (CIBERehd), 28029 Madrid, Spain; 4Medicine Department, University of Zaragoza, 50009 Zaragoza, Spain

**Keywords:** *H. pylori* infection, management, antibiotic resistance, primary care services

## Abstract

*Helicobacter pylori* (*H. pylori*) is a key agent in several upper gastrointestinal diseases. Treatment of *H. pylori* infection is the main strategy for resolving the associated gastroduodenal damage in infected patients and for the prevention of gastric cancer development. Infection management is becoming complex due to the increase in antibiotic resistance, which already represents a global healthcare problem. Resistance to clarithromycin, levofloxacin or metronidazole have forced the adaptation of eradication regimens in this new reality to reach the eradication rate target recommended in most international guidelines (>90%). In this challenging scenario, molecular methods are revolutionizing the diagnosis of antibiotic-resistant infections and the detection of antibiotic resistance and opening a path towards personalized treatments, although their use is not yet widespread. Moreover, the infection management by physicians is still not adequate, which contributes to aggravating the problem. Both gastroenterologists and mainly primary care physicians (PCPs), who currently routinely manage this infection, perform suboptimal management of the diagnosis and treatment of *H. pylori* infection by not following the current consensus recommendations. In order to improve *H. pylori* infection management and to increase PCPs’ compliance with guidelines, some strategies have been evaluated with satisfactory results, but it is still necessary to design and evaluate new different approaches.

## 1. Introduction

*Helicobacter pylori* (*H. pylori*) is a Gram-negative bacteria that infects the gastric epithelium. It was discovered in 1983 by Barry J. Marshall and J. Robin Warren, and its discovery introduced a complete revolution in gastroenterology [1]. Since then, *H. pylori* has been a key agent in multiple upper gastrointestinal digestive diseases. *H. pylori* infection has been associated with benign diseases, such as chronic gastritis, gastric intestinal metaplasia, gastric and duodenal peptic ulcers or dyspepsia, as well as malignant diseases (gastric carcinoma or mucosa-associated lymphoid tissue (MALT) lymphoma) [2].

Despite its relatively recent discovery, researchers suggest that it has lived with humans for 88,000–116,000 years [3] and is distributed throughout the world. *H. pylori* infection affects almost half of the world’s population, but the prevalence among different regions is heterogeneous. The prevalence of *H. pylori* infection has been declining in highly industrialized countries, whereas the prevalence has stabilized at a high level in developing and newly industrialized countries [4,5,6].

Two different transmission mechanisms have been described, oral–oral or fecal–oral transmission. Oral–oral transmission appears to be the main route of transmission in *H. pylori* infection. This route would explain the frequent intra-family infection transmission. Fecal–oral transmission is the second transmission mechanism described; it occurs mainly through contaminated water when sanitary conditions are inadequate. [7].

*H. pylori* infection eradication is the key method of treating upper gastrointestinal lesions associated to the infection and for preventing gastric cancer development [8]. The Kyoto *H. pylori* consensus conference in 2015 recommended that all *H. pylori* infections should be eradicated (except for major reasons, such as the presence of important comorbidities) [9].

In order to improve *H. pylori* infection management, multiple national and international consensus documents have been developed to improve its diagnosis and therapy. These recommendations are updated periodically with the latest scientific evidence [10,11,12].

One of the most current worries is the global increase in antibiotic resistance associated with *H. pylori* infection. Furthermore, today, *H. pylori* infection is managed almost exclusively by primary care physicians (PCPs), who are not always aware of the most recent recommendations for the management of the infection, or those recommendations take longer to be implemented at that level (references). Therefore, there is a need to develop strategies to reduce the gap between the different levels of *H. pylori* management.

The aim of this review is to analyze different factors in the diagnosis and treatment of *H. pylori* infection in real practice, such as currently available diagnostic methods for *H. pylori* infection and future perspectives, and the worrying increase in antibiotic resistance and its influence on guideline recommendations for prescribing different eradication regimens for *H. pylori* infection. On the other hand, we analyze the diagnosis and treatment carried out in real practice by the physicians who usually managed this infection, such as gastroenterologists and mainly PCPs, as well as different strategies to improve the diagnosis and treatment of *H. pylori* infection.

## 2. Diagnosis of *Helicobacter pylori* Infection and Antibiotic Resistances Determination

Diagnosis of *H. pylori* infection is one of the most important step in infection management. Several diagnostic methods are currently available for detecting *H. pylori* infection, with both high sensitivity and specificity. Diagnostic tests are classified into non-invasive and invasive (when an endoscopy is needed) methods. The non-invasive diagnostic tests are the urea breath test, stool antigen test, serological tests and tests using molecular methods. On the other hand, the invasive diagnostic available methods are endoscopic imaging, histology determination, rapid urease testing, and tests using culture and molecular methods. Each method has advantages, disadvantages and limitations (Table 1). The choice of the diagnostic method is multifactorial (sensitivity and specificity in different scenarios, patient preference and characteristics and finally the availability of diagnostic tests) [13].

### 2.1. Non-Invasive Tests

Non-invasive tests are those that do not require an endoscopy to diagnose *H. pylori* infection. They have some advantages because they are cheaper, safer and easier for the patient. Performing a gastroscopy is an invasive procedure, and some complications may occur from sedation or the procedure itself. On the other hand, in patients at risk or even with comorbidities, endoscopy may be contraindicated. In addition, this procedure can only be performed in hospitals, so its availability and accessibility are more limited. To solve these problems, non-invasive tests have been developed. However, they also present limitations, as is not possible to perform some techniques, such as cultures or diagnosis molecular methods [13].

#### 2.1.1. Urea Breath Test

The urea breath test (UBT) is based on a particular mechanism of *H.pylori* (urease enzyme activity). First, the patient swallows a ^13^C- or ^14^C-labeled urea tablet. Subsequently, due to the urease present in *H.pylori*, the urea is broken down, labeled CO_2_ is released, and then it is absorbed in the blood. Finally, labeled CO_2_ is exhaled, thus allowing the measurement of its concentration. The UBT has a high sensitivity and specificity close to 95% [13]. To reach this accuracy, it is necessary to follow a protocol, and patients must discontinue PPI two weeks and antibiotics at least four weeks before the test in order to decrease false negative results. The Maastricht VI consensus also indicates that the use of citric acid helps to slow gastric emptying and enhance the gastric distribution of the ^13^C or ^14^C urea, which increase its contact time with *H. pylori* urease [11].

Nowadays, UBT is still one of the most commonly used tests due to its characteristics (low cost, high availability and the ease of performing it). UBT remains an important diagnostic method in *H. pylori* infection before and after eradication treatment [11].

#### 2.1.2. Stool Antigen Test

The stool antigen test (SAT) is other non-invasive test with high sensitivity and specificity. This diagnosis method analyzes and detects the presence of the *H. pylori* antigen in fecal samples. *H. pylori* strains in the stomach can produce bacterial antigens, which are execrated in the patients’ stool. Different tests can be used to detect those *H. pylori*-specific antigens in stool samples. The first developed kits detected antigen with polyclonal antibodies, but more recently developed tests that use monoclonal antibodies have higher accuracy [11].

There are two types of SAT based on different technologies: the enzyme immunoassay (EIA) and rapid immunochromatography assay (ICA). EIA test kits need a laboratory to analyze the sample and are better than the rapid test. The latest ones have more availability and can be used by primary care physicians or directly by the patient themself, but they have low sensitivity and specificity, and users must be warned of these limitations [13]. The recent international consensus Maastricht VI recommended using monoclonal SATs before and after *H. pylori* treatment if the kits are properly validated [11].

#### 2.1.3. Antibody-Based Test

Detection of IgG antibodies against *H. pylori* is another test for the diagnosis of *H. pylori* infection. There are two types of test similar to SAT, EIA and rapid ICA, but EIA is the most common technique [13]. The main problem with serology is that tests can be positive many months after eradication, because antibodies remain in blood for a very long period. Therefore, antibody-based tests are not a valid alternative for post-eradication confirmation [13].

Another important limitation is that tests need to be locally validated. Due to the huge variability of *H. pylori* bacteria, some tests could not be valid for all countries or regions [11].

The recent international consensus indicates some specific situations in which serological tests can be used to determine the presence of (current or past) infection, including benign pathologies such as bleeding ulcers or malignant pathologies (gastric cancer or gastric MALT lymphoma), and other special situations such as recent use of PPI or antibiotics [11].

### 2.2. Invasive Tests

All invasive tests are based in the performance of upper GI endoscopy. Endoscopy images allow detecting mucosal features that could suggest the presence of *H. pylori* infection. Those mucosal features are not specific enough for confirming a *H. pylori* diagnosis. Other methods based on conventional endoscopy have been developed, such as chromoendoscopy or, more recently, magnifying endoscopy that provides direct observation of gastric mucosa microstructure, in which histopathological changes produced by *H. pylori* infection can be detected with high sensitivity and specificity in the corpus, but lower sensitivity and specificity in the antrum of the stomach. Those differences limit the use of magnifying endoscopy in real clinical practice [13].

The main advantage of endoscopy is that it permits obtaining gastric mucosal samples from biopsy that can be used for other studies, including the rapid urease test, histology, culture and molecular methods.

#### 2.2.1. Histology

Histology is considered the gold standard in the direct diagnosis of *H. pylori* infection, but some factors could be important for diagnosis. Diagnostic accuracy may be influenced by the experience of the pathologist, the number and place of collection of biopsies, or the use of antibiotics or PPI.

PPI should be discontinued two weeks before performing endoscopy, and biopsies should be obtained from the antrum and the corpus (two from each location) and submitted in different containers. An additional biopsy obtained from the incisura allows determining the gastritis histological staging OLGA (Operative Link on Gastric Atrophy) and OLGIM (Operative Link on Gastric Intestinal Metaplasia), which ranks the patient’s cancer risk and determines the follow-up endoscopies [11].

Many different stains have been used to detect *H. pylori* in gastric samples. The hematoxylin–eosin (HE) stain, Giemsa, Warthine–Starry, and silver stain are frequently used, although HE is most commonly reported one used in clinical practice. However, the most sensitive and specific stain is the immunohistochemical stain [13].

#### 2.2.2. Rapid Urease Test

Rapid urease tests are very common because they are easy to perform and cheap. These tests are based on *H. pylori* urease enzyme activity. The presence of the bacteria in biopsy samples produces ammonia from the urea of the kit, which increases the pH and subsequently changes the color on the pH monitor. Available commercial kits have a high sensitivity and specificity (95 and 85%, respectively), but it is necessary to take into account many circumstances that could produce false negative results, such as the density of bacteria in the biopsy (10,000 bacteria are required for a positive result), the use of PPI, antibiotics, bismuth, H2 receptor antagonists, the presence of blood or reading the test earlier than recommended [13].

#### 2.2.3. Culture

Culture of *H. pylori* from a gastric biopsy is the method with the highest specificity, which is close to 100% specificity, but the sensitivity is lower (85–95%) because the transport and culture of *H. pylori* is highly complex [13].

Although culture is not very accessible since it is not performed in all hospitals, and it is expensive and laborious, the main advantage of this technique is that it provides the antibiotic sensitivity of *H. pylori.*

The determination of antibiotic susceptibility can help guide specific eradication regimens after the failure of one or multiple treatment lines. Culture is an adequate diagnostic method to detect antibiotic resistance in patients with multiple therapeutic failures. It can also be used at the population level to detect antibiotic resistance in a specific geographical area. [8,11].

#### 2.2.4. Molecular Methods

The use of molecular diagnostic methods such as polymerase chain reaction (PCR) has been employed for the diagnosis of *H. pylori* infection. These methods have been applied in several human samples (saliva, fecal samples or gastric biopsies).

Molecular methods that are PCR-based have high sensitivity and specificity (more than 95%), and multiple genes have been used for the diagnosis of *H. pylori infection* (16S rRNA, 23S rRNA, UreA, glmM, UreC, HSP60 or VacA). The use of two genes in the same test may increase diagnostic accuracy mainly when non-gastric samples are used. This technique has many advantages such as faster results, no need for specific and complex transportation methods, and a lower bacterial load in the sample for a positive result [13].

In addition to the diagnosis of *H. pylori* infection, this technique analyzes antibiotic susceptibility. There are multiple tests for the detection of clarithromycin, levofloxacin and tetracycline antibiotic resistance, but the diagnosis of antibiotic resistance to amoxicillin and metronidazole is still difficult due to the multiple biological mechanisms involved in these forms of antibiotic resistance development [8].

For the first time, the last international Maastricht VI consensus recommended PCR-based clarithromycin susceptibility testing before prescribing therapy with eradication regimens with clarithromycin because of the dramatic increasing prevalence of antibiotic resistance, especially clarithromycin resistance, in some world regions [11].

The main limitation of this technique is that even though it has been used in multiple types of samples, the best results are obtained with biopsies of the gastric mucosa, so it is still necessary to perform an endoscopy to obtain the sample [8,11]. In this field, very promising diagnostic tests are currently being developed that allow the detection of antibiotic resistance in fecal samples, thus enabling the diagnosis of antibiotic resistance in non-invasive samples [14]. For example, in 2018, a Chinese group described a method to simultaneously detect the clarithromycin 23sRNA gene in gastric in fecal samples using droplet digital PCR (ddPCR) with promising results [15]. Another Italian-American group published that another PCR-based assay of stool samples can be used to detect mutation associated with clarithromycin resistance at a high sensitivity rate of 93.6% [16].

The future of molecular methods lies in the improvement of DNA extraction and amplification applied to stool samples and overcoming the need to perform an upper GI endoscopy to obtain mucosal samples. On the other hand, it is necessary to reduce the cost of the techniques and increase their availability in order to extend their use and carry out targeted antibiotic therapies [8].

## 3. *Helicobacter pylori* Infection Treatment and Antibiotic Resistances

Historically, treatment of *H. pylori* infection is more difficult than treatment of other bacterial infections due to the particular characteristics of this infection.

On the other hand, *H. pylori* antibiotic resistance has been increasing over the years, with a subsequent decrease in successful eradication rates. This is a multifactorial problem with many points of view.

First of all, the limited efficacy of antibiotics reduces the therapeutic options. There are only a few antibiotics that can be used in clinical practise with acceptable levels of effectiveness (i.e., amoxicillin, clarithromycin, metronidazole, tetracycline, levofloxacin and rifabutin), and they must always be administered in combination therapy comprising at least two or three antibiotics [8]

Another point related to the increase in antibiotic resistance is the exceptional adaptation ability of *H. pylori*. A high number of resistance mechanisms have been described, including gene mutation, and physiological changes, such as biofilm, coccoid formation or changes in antibiotic uptake and efflux.

Three profiles of resistance have been described: single drug resistance, multidrug resistance and hetero-resistance.

This last mechanism, which has important clinical implications, is due to heterogeneous populations of *H. pylori* that exhibit different antibiotic resistances in the same host due to (1) co-infection with strains with different antibiotic resistance profiles, (2) the spontaneous emergence of resistant strains or, more commonly, (3) the selection of resistant strains from the bacterial population as a result of antibiotic pressure that generates resistant subpopulations [8].

Hetero-resistance of *H. pylori* is an aspect of antibiotic resistance that is usually overlooked in consensus documents of *H. pylori* infection management, but it is a relevant point to take into account in the antibiotic resistance development of *H. pylori* infection [8]. A recent meta-analysis indicates that hetero-resistance in developing country populations is due to simultaneous infection by multiple *H. pylori* strains. On the other hand, in developed countries, the reason for the increase in hetero-resistance is explained as the selection of resistant strains due to high antibiotic pressure [17]. Different studies showed that antrum and corpus mucosa have different characteristics, which implies a natural selection mechanism for for *H. pylori* strains [8]. This could explain data from mentioned meta-analysis, which showed a discrepancy of antibiotic susceptibility tests on strains isolated from different anatomical sites of the stomach [17]. So, if only biopsies were taken from one site, strains with antibiotic resistance could be underestimated. Finally, prevalent hetero-resistant *H. pylori* strains are frequently associated with the antibiotics prescribed in *H. pylori* eradication regimens (60.1% to clarithromycin, 61.1% to metronidazole, 46.1% to levofloxacin or 3.8% to amoxicillin) [17]. Therefore, hetero-resistance needs to be taken into account in future treatment guidelines.

Finally, the most important cause of increasing antibiotic resistance is the extended use (and in many cases, overuse and misuse) of some antibiotics for other infections, mainly macrolides (clarithromycin and azithromycin), which are prescribed for respiratory, genital or urinary infections. According to prior studies, antibiotic exposure is so widespread that 80% of patients prescribed an eradication treatment had previously received amoxicillin, while 46% and 11% had taken macrolides or quinolones, respectively [18,19]. In another study, global macrolide and quinolone consumption increased by 19% and 64%, respectively, from 2000 to 2010 [20].

A recent study evaluated antibiotic resistance to *H. pylori* in a huge population of naïve patients for *H. pylori* eradication treatment. The study described that only 49% of patients had no antibiotic resistance in 2013, whereas this rate decreased to 36% in 2020. The study also showed antibiotic resistance rates to clarithromycin, metronidazole and levofloxacin of 25%, 30% and 20%, respectively [21].

When comparing the rates of antibiotic resistance in patients who had undergone one eradication treatment (non-naïve), an increase in resistance to any of the evaluated antibiotics was observed; 66%, 54% and 28% resistance rates to clarithromycin, metronidazole and levofloxacin were reported. Other worrying data are the increased rate of dual resistance to both clarithromycin and metronidazole and triple resistance to clarithromycin, metronidazole and levofloxacin, with rates ranging from 13% and 6%, respectively, in naïve patients to 43% and 19%, respectively, in non-naïve patients [21].

This problematic situation of increasing antibiotic resistance and decreasing eradication rates of *H. pylori* infection led the World Health Organization (WHO) in 2017 to include *H. pylori* as a high priority on the list of the 20 pathogens that may pose the greatest threat to health due to drug resistance [22].

Antibiotic resistance is not homogeneous around the world—it varies from one continent to another and even between regions in the same geographical area. Accordingly, a marked difference was described between southern and northern Europe. Overall, *H. pylori* resistance in northern European countries was lower than that in southern Europe (31.5% vs. 56% respectively). Resistance in southern Europe was greater for clarithromycin and levofloxacin (28% and 23.5%, respectively), as opposed to northern Europe, with rates below 10% for both antibiotics. Mostly, the same trend was observed for dual and triple resistances in southern Europe (15% and 7.5%, respectively) versus northern Europe (3.5% and 0.3%, respectively) [21].

In the Asia Pacific region, there is a huge variability between countries. Compared to Europe, a higher levofloxacin resistance has been shown in Korea (44% in 2013 and 62% in 2017), and resistance rates for metronidazole, amoxicillin and tetracycline were similar [23,24]. In Latin America, clarithromycin, metronidazole and levofloxacin resistances of 13%, 50% and 19%, respectively, were reported [25]. In the United States (USA), there were also differences in drug resistance patterns, with a lower clarithromycin resistance (17%), but higher metronidazole and levofloxacin resistances (43.6% and 57.8%, respectively) compared to Europe [26].

Other effective antibiotics against *H. pylori* infection usually have lower resistance rates, but there are also significant differences between geographical areas. Resistance rates to amoxicillin and tetracycline in Europe are very low (less than 1%), but in the USA, they are higher (6.4% and 2.8%, respectively) and similar to Latin America. The highest resistance rates were observed in Asia, where the amoxicillin resistance rate varied from 6% to 8% and the tetracycline resistance rate from 4% to 16% [21,23,24,25,26].

The problem of drug resistance to *H. pylori* is one of the main causes for the sequential adaptation and updates of both national and international consensus documents on *H. pylori* infection.

In order to achieve the goal of any *H. pylori* antimicrobial therapy, meaning an eradication rate over 90%, the last European consensus document published (Maastricht VI/Florence consensus report) recommends analyzing the individual’s antibiotic susceptibility or, if not possible, determining the population antimicrobial resistance rate regularly to modify antibiotic regimens [11].

In this consensus document, eradication treatments differed depending on whether patients were located in a low-clarithromycin-resistance area (less than 15%) or a high-resistance area (more than 15%) (Figure 1). In high-resistance areas, bismuth quadruple therapy (proton pump inhibitor (PPI), bismuth, tetracycline and metronidazole) or quadruple levofloxacin therapy (PPI, levofloxacin, amoxicillin and bismuth) are recommended as first-line treatments, whereas in the very few areas remaining with low-clarithromycin-resistance rates, clarithromycin triple regimens (PPI, clarithromycin and amoxicillin) or bismuth quadruple regimens could be effective in order to achieve the eradication rate goal. Currently, even the prevalence of dual resistance (to clarithromycin and metronidazole) in high-resistance areas is taken into account. A recent review suggests that concomitant therapy (PPI, clarithromycin, amoxicillin and metronidazole), which is a common treatment regimen in many countries as a first-line treatment, should not be used if the prevalence of dual resistances is more than 15% [27].

Bismuth quadruple: proton pump inhibitor (PPI); bismuth, tetracycline and metronidazole. Levofloxacin quadruple: PPI, levofloxacin, amoxicillin and bismuth. Levofloxacin triple: identical eradication regimen, but without bismuth. Clarithromycin triple: PPI, clarithromycin and amoxicillin; only used if clarithromycin sensitivity is known or if it is an effective eradication regimen in the specific geographical area. Non-bismuth quadruple (concomitant): PPI, clarithromycin, amoxicillin and metronidazole. In geographical areas with high fluoroquinolone resistance (>15%), the combination of bismuth with other antibiotics, high-dose dual PPI–amoxicillin or rifabutin, could be an alternative treatment.

Another consequence of high resistance rates is that treatment duration has increased in order to achieve optimal eradication rates. The European consensus recommends 14 days of treatment for a concomitant regimen (PPI, amoxicillin, clarithromycin and metronidazole) instead of 10 days. A study from Hp-EuReg (European registry on the management of *Helicobacter pylori* infection) shows a higher eradication rate in 14-day regimens compared to 10-day regimens [28]. Another real-life study comparing 10-day versus 14-day regimens demonstrated similar data, with higher eradication rates in the 14-day group (96.1% vs. 80%) [29]. A Spanish study recently published by our group showed higher eradication rates for longer treatment regimens, with a 56.7% rate for 7 days, 72.6% for 10 days and 82.3% for 14 days [30].

Other treatment regimens are following the same trend of increased duration. The Maastricht VI consensus recommends continuing bismuth quadruple therapy (PPI, bismuth, tetracycline and metronidazole) for 14 days (10-day therapy can also be considered if it presents the same effectiveness), although more studies should be conducted to better define the optimum treatment duration in populations for which the resistance pattern is known [11].

In conclusion, the current increase in antibiotic resistance is an important problem in *H. pylori* infection management, which determines the need to use a greater number of antibiotics in eradication regimens, to not use certain drugs in areas with a high rate of resistance and to carry out longer antibiotic treatment.

## 4. Diagnosis and Treatment of *H. pylori* Infection by Gastroenterologists in Real Clinical Practice

The adaptation of daily clinical practice to current guidelines’ recommendations is a very important point in the approach to *H. pylori* infection management. The request for a *H. pylori* infection diagnosis for conditions that are not recommended in the guidelines and the use of ineffective antibiotic treatments can worsen the problem of increased antibiotic resistance. Due to all these circumstances, it is essential to incorporate the current guidelines into daily care activities as soon as possible, even if this implies a process of continuous updating of *H. pylori* infection management.

Some studies have assessed the management of *H. pylori* infection, mainly by gastroenterologists who, until a few years ago, were those who mainly managed *H. pylori* infection. A study published by our group showed that *H. pylori* diagnosis and treatment by a gastroenterologist was suboptimal. It showed a 7.2% rate of inappropriate requests for a urea breath test to investigate *H. pylori* infection, according to national guidelines [31,32]. The most frequent inappropriate indications were gluten intolerance, heartburn and diarrhea. Gastroenterologist treatment prescriptions were also suboptimal. Only 73.9% of the treatment regimens prescribed were considered appropriate, which was linked to low eradication rates (81.4%) [33].

Another study by the Hp-EuReg group concluded that *H. pylori* infection management by European gastroenterologists needs to improve. *H. pylori* infection management was extremely heterogeneous, with strong regional differences. The triple therapies not recommended in most countries because of the increase in antibiotic resistance. These therapies are in a state of disuse in southern Europe and their use is declining in Eastern Europe, but they are still used in northern Europe. This use of inadequate treatment regimens causes a high rate of failures and low eradication rates (63.2–86.6% in patients who receive triple therapy). These data could be the result of an incomplete incorporation of the last consensus documents’ statements, which recommended changing from triple to quadruple therapies. Globally, triple therapies decreased from 50% to 32% between 2013 and 2018 in parallel with increasing eradication rates, which are now close to a 90% cure rate [28]. These data are similar to that of a Chinese study based on personal surveys. Up to 40% of the respondents did not follow diagnosis recommendations, less than 70% prescribed adequate eradication regimens and 20% of respondents did not confirm eradication after antibiotic treatment [34].

Due to this incomplete penetration of the clinical practice guidelines, some studies have investigated the most common mistakes in clinical practice among gastroenterologists. A European study found that the most repeated mistakes were the use of the standard triple therapy, the prescription of eradication therapy for only 7 to 10 days, the use of a low dose of PPI, the recurrence of the prescription of the same antibiotics after an eradication failure and the absence of confirmation of eradication success. However, there are also positive data because time-trend analyses showed progressively greater compliance with guidelines [35].

Therefore, we can conclude that the management of the infection by gastroenterologists needs to improve attempts to achieve a greater penetration of the guidelines in order to increase eradication rates and achieve the target set at >90%. Nevertheless, a progressive trend towards better management of the infection must be acknowledged.

## 5. The New Role of Primary Care Services in Diagnosis and Treatment of *H. pylori* Infection

Nowadays, dyspepsia is the main symptom for testing *H. pylori* infection. There are a significant number of patients with this condition; according to certain research, 20–45% of the population has dyspepsia symptoms [36], making it a common reason for medical consultations. Of the 54 million outpatient visits to gastroenterologists in the United States, 21.8 million were for abdominal pain, and in over 3 million cases, dyspepsia was the final diagnosis [37].

Primary care physicians (PCPs) are often accustomed to the management of dyspepsia in everyday clinical practice. Approximately 5–7% of patients at primary care-level consults have symptoms that are related to dyspepsia [38,39].

*H. pylori* infection management has ceased to be a major task for gastroenterologists due to the large number of patients with dyspepsia symptoms and the confirmation of the test-and-treat strategy for non-investigated dyspepsia as a valid method in the infection management [11]. This strategy refers to non-invasive testing for H. pylori in patients with dyspeptic symptoms and to eradication of the infection whenever detected [11]. As a result, primary care levels have become more significant in managing *H. pylori* infection in recent years, and currently, PCPs have been placed on the front-line of *H. pylori* management.

In a Spanish study conducted among PCPs, a total of 87.7% of respondents reported they had indicated eradication treatment at least once during the previous years, and 32.8% had prescribed more than 6 eradication regimens [40]. Similar results were observed in other studies reported in Spain, Israel and Croatia, where *H. pylori* infection management performed by PCPs was also very common [41,42,43,44].

The question now is to determine whether PCPs manage *H. pylori* infection at least in a similar way to that reported by the specialists in order to detect mistakes and introduce measures to correct them.

Some studies that have assessed the management of *H. pylori* infection specifically in Primary Care Services have shown suboptimal management of the infection at this level. The management is not adequate at different levels, including the indication of diagnosis and treatment of the infection by prescribing inappropriate antibiotic treatments with consequently lower eradication rates. A Spanish study based on personal surveys to PCPs reported poor adherence to guidelines in 2.8%, a moderate adherence in 77.5% and only an optimal adherence in 19.7% of PCPs [41].

The penetration of guidelines concerning the diagnosis of *H. pylori* infection at the primary care level is not appropriate, and therefore, its management is suboptimal. In an Israelite study focused only in PCPs, *H. pylori* was diagnosed in most cases by the C13-urea breath (UBT) test or fecal antigen test, but 2.8% of the PCPs used serology, which is not recommended by guidelines [42]. A 2008 Spanish study found similar data, with a high percentage (12.7%) of inadequate non-invasive tests used for the diagnosis of *H. pylori* infection [40]. The guidelines recommendations for the discontinuation of antibiotics and PPI 4–6 weeks and 14 days before UBT [11], respectively, were only followed by 40.9% of Israelite PCPs [42], similar to the data of a Croatian study based on PCP surveys [43]. In the second part of the Israeli study, a clear positive time trend was observed in the correct diagnosis of active *H. pylori* infection, with progressively lower use of serology and greater following of the adequate recommendations of PPI and antibiotic interruption before UBT, which increased from 40% to more than 60% [44].

An important step in *H. pylori* diagnosis is the indication to request a *H. pylori* diagnosis test, because *H. pylori* screening is only cost-effective in populations with an intermediate or high incidence of gastric cancer (higher than 15–20 per 100.000), something that cannot be implemented in western countries, where the incidence of gastric cancer is low [45].

In a study published by our group, the inappropriate indication rate (Table 2) was high (35.9%). The most frequent inappropriate request for *H. pylori* diagnosis was gastroesophageal reflux (GERD), following by non-specific abdominal pain [33]. Another Spanish study reported that 18.3% of antibiotic regimens were prescribed for GERD [40]. These data are similar to those reported in other countries. In a Croatian study, a high proportion of PCPs (43.0%) tested for *H. pylori* infection in patients with GERD symptoms [43]. In an Israeli study, 83% of PCPs tested for *H. pylori* infection consistently for this pathology [42]. In a Mexican study, 41.8% of PCPs reported that GERD was an indication for *H. pylori* infection [46]. However, according to international consensus documents, there is no current evidence to test and treat *H. pylori* infection in patients with GERD [11].

At the treatment level, suboptimal management has also been reported by different studies, which demonstrated a partial penetration of current guidelines. Current guidelines do not recommend triple therapies as first-line eradication regimens because of the high resistance rates to clarithromycin in most countries, which has been associated with low eradication rates. However, triple therapy is still more frequently prescribed at the primary care level than by gastroenterologists.

In a study conducted in Israel in 2016, 93% of PCPs surveyed prescribed triple therapies as the first-line treatment [42]; this percentage decreased to 62.9% in 2018 after an educational intervention [44]. In a Croatian study, the data were similar, and 66.3% of PCPs prescribed triple therapy as the first treatment line [43]. Data from Spain were similar; in a study based on PCPs surveys, the most commonly used treatment of choice was triple therapy (56.4%) [41], similar to a study conducted by our group, in which triple therapy was prescribed in 63.1% of cases [33]. However, the positive data were that these inappropriate eradication regimens decreased to 18.5% after an educational intervention and to 4.8% for PCPs who received a second educational intervention [33,47].

Data from other continents also demonstrate inappropriate treatment regimen prescription. In Mexico, the most widely used antibiotic regimen was clarithromycin triple therapy (63.8%) [46].

The last important step in the management of *H. pylori* is the control of eradication of the infection, which is mandatory, according to international guidelines [11]. After *H. pylori* eradication treatment, 67.6% of PCPs in Spain always referred to a confirmatory diagnostic method, but 27.7% did so depending on the clinical situation, and 4.7% never tested [41].

Another Spanish study founded similar data. After an eradication regimen, PCPs requested another UBT in 72.4% of patient. Unfortunately, training sessions delivered by gastroenterologists to PCPs to improve *H. pylori* infection management did not improve this goal (69% in primary care centers with training sessions vs. 74.4% in primary care centers without training sessions) [47]. Other studies from different countries reported even worse results. Only 64.7% of Croatian PCPs requested an UBT after an eradication treatment (64.7%) [43], and 43.6% of Israeli PCPs routinely confirmed eradication [42], similar to data published in Ireland (48%) [48]. Studies from Pakistan (12%) [49] and Korea (9.3%) reported the lowest rates for this aspect [50].

Therefore, as we can observe, *H. pylori* infection management is still suboptimal at the primary care level. This could be due to multiple factors. First, the field of action of PCPs is very broad, and although the prescription of antibiotic treatments for *H. pylori* eradication is frequent in clinical practice, it is a minor task in PCPs’ activities. In a study, 94% of PCPs prescribed less than 10 eradication treatments per month [41]. The second aspect is that clinical practice guidelines are mostly developed by expert gastroenterologists, and new changes and updates penetrate more slowly at the primary care level (56.2% of Spanish PCPs questioned in a national survey have never read the European consensus document). Furthermore, local antibiotic resistances that condition the success of eradication treatment are usually unknown. Only a minority of doctors responded that they knew the local resistance rate to clarithromycin (13.0%) in Spain [41]. Physicians who claimed to not know these figures underestimated the real resistance rate. The last part of this multifactorial problem is the lack of specific training programs for *H. pylori* infection management by PCPs. Only 33.5% of PCPs claimed to have received a specific course of *H. pylori* infection management or dyspepsia management during their career, and just 27.4% claimed to have received one in the last 5 years [41].

For all these reasons, strategies to improve *H. pylori* infection management at the primary care level must be carried out to achieve better adherence to current guidelines in daily clinical practice.

## 6. Strategies to Improve *H. pylori* Infection Diagnosis and Treatment—A Long Way to Go

Strategies to improve *H. pylori* infection diagnosis and treatment and, as a consequence, to reduce antibiotic resistance can be grouped into three areas, which include the development of new tools to detect antibiotic resistances for each patient, performing strategies to increase doctors’ compliance to current guidelines, and strategies to improve therapeutic compliance by patients.
First of all, it is important to accurately investigate local antibiotic resistance, as well as the eradication rates of the different antibiotic regimens, in order to adapt the guidelines for each region. One step forward beyond the investigation of any specific population’s antibiotic resistance is to detect the antibiotic resistance to *H. pylori* infection in each patient before they receive any eradication treatment. Traditionally, the gold standard to determine antibiotic resistance was *H. pylori* culture, but this technique is not very accessible since it is not performed in all hospitals and, consequently, cannot be requested at all primary health centers. This method is slow, expensive and challenging, and it requires an upper GI endoscopy to obtain samples. Nowadays, several methods for the rapid detection of resistance have been developed based on the detection of specific *H. pylori* gene mutations encoding resistance. Those PCR-based molecular techniques are quicker than culture and offer some cost-efficient advantages, but it is still expensive to apply them at the population level. There are reliable kits to detect resistance to clarithromycin, tetracycline and levofloxacin. In contrast, diagnosis of antibiotic resistance to amoxicillin and metronidazole is still difficult due to the multiple biological mechanisms by which these forms of antibiotic resistance are developed [8].Another problem of those PCR-based molecular techniques is that, although they can be applied to a large number of samples, they currently only have adequate sensitivity and specificity in gastric samples that require endoscopy. Therefore, future research on PCR techniques based on next-generation sequencing technologies should be aimed at being able to detect a greater number of mutations for the whole bacterial resistome. Finally, those techniques should be able to be applied reliably in non-invasive samples, such as stool samples, to apply them at the population level [8,14].To improve *H. pylori* infection management, it is essential to develop new strategies in order to adapt national and international consensus documents to daily clinical practice. Several studies have demonstrated that infection management is suboptimal in both gastroenterologists and PCPs, but very few studies have evaluated strategies to improve it, especially at the primary care level, where a large number of eradication treatments are now being prescribed.In a study from Israel, PCPs received three interventions. Firstly, printed materials were distributed to PCPs. These previously printed materials summarized and explained the implications of *H. pylori* infection and antibiotic regimens based on current guidelines. Secondly, trained personnel carried out an educational visit. During the visits, previously sent written materials were discussed. Finally, a virtual platform was developed through which PCPs could interact with a gastroenterologist who answered their questions in less than an hour. The platform made it possible to respond to comments and for the rest of the participants to see them. [44]. After those interventions, *H. pylori* infection first-line treatment compliance with the guidelines improved, mainly in the group who used the social media platform.Our group evaluated two different strategies in a large group of PCPs. Firstly, we evaluated the implementation of sending written recommendations (indications of *H. pylori* diagnosis and antibiotic treatments according to current guidelines). This intervention improved the use of adequate antibiotic treatments and had an impact in improving the infection eradication rates at Primary Care Level (it improved from 63.7% to 81.4%). On the other hand, this intervention did not improve indications for the *H. pylori* diagnosis [33].In a second study, we evaluated the effect of sending written recommendations based on current national consensus documents to PCPs, followed by group oral training sessions, compared to sending those written recommendations alone. Giving training sessions was an effective strategy to improve the appropriateness of treatment prescriptions compared to providing just written recommendations only. However, this strategy did not improve the appropriateness of indications for UBT requests or eradication rates [47].Finally, some studies have evaluated patient-based strategies. A prospective clinical trial in China demonstrated an improvement in eradication rates via sending text messages. On the other hand, another Chinese retrospective study did not demonstrate an improvement in eradication rates via the use of patient–doctor text messages. Finally, a meta-analysis did show that patient education positively influences therapeutic adherence and eradication rates. [51,52,53].

## 7. Conclusions

In conclusion, many strategies to improve *H. pylori* infection diagnosis and treatment have been developed and are successful in most cases. However, the diagnosis and management of the infection are still suboptimal, and it is necessary to continue developing new strategies for their improvement at all levels. The progressive increase in antibiotic resistances represents a real and serious health problem. The improvement and development of diagnostic methods, such as PCR-based molecular techniques, can be promising tools to detect antibiotic resistance at an individual level and, thus, be able to offer targeted antibiotic treatment. Another strategy must be to achieve high levels of appropriate *H. pylori* diagnosis and treatment at both the GI specialist and PCP levels.

## Figures and Tables

**Figure 1 diagnostics-13-01918-f001:**
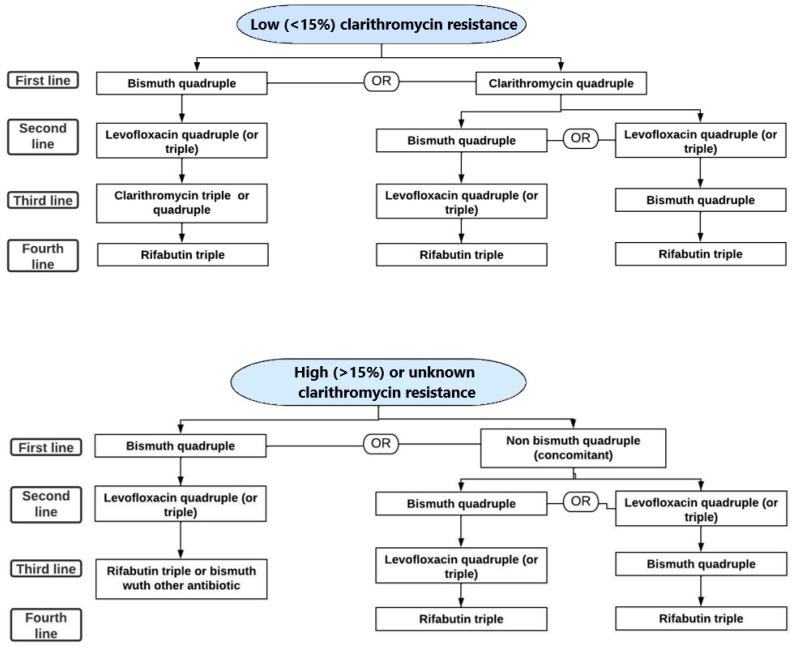
Algorithm for empirical *Helicobacter pylori* (*H. pylori*) eradication if individual antibiotic susceptibility testing is not available (Adapted from the Maastricht VI/Florence consensus report). REFERENCE 11

**Table 1 diagnostics-13-01918-t001:** *H. pylori* infection diagnostic methods and their advantages and disadvantages.

*H. pylori* Infection Diagnostic Methods	Advantages	Disadvantages
Urea breath test	- Low cost - Easy to perform - High sensitivity and specificity - Could be used for diagnostic and post-eradication confirmation	- Patients must discontinue PPI and antibiotics before the test
Stool antigen test	- Easy to perform - Could be used for diagnostic and post-eradication confirmation - EIA-based method: High sensitivity and specificity - ICA-based method: - More availability - Could be used by PCPs or by the patient themself	- EIA-based method: Need a laboratory to analyze the sample - ICA-based method: Low sensitivity and specificity
Antibody-based test	- Valid for some specific situations in which other diagnostic methods could not be used, such as bleeding peptic ulcers, gastric mucosal atrophy and recent use of antibiotics or PPI	- Not valid for post-eradication confirmation - Commercial tests need to be locally validated
Endoscopic image	- Allows evaluating mucosal features	- Not specific enough to confirm *H. pylori* diagnosis
Histology	- Allows determining gastritis histological staging (gastric atrophy and intestinal metaplasia), which ranks the patient cancer risk	- Several factors influence the diagnostic accuracy, such as the site, number of biopsies, PPI use, antibiotics use and pathologist experience - Patients should discontinue PPI before the test
Rapid urease test	- Low cost - Easy to perform	- Many circumstances could produce false negative results, such as the density of bacteria in the biopsy, use of PPI, antibiotics, bismuth, H2 receptor antagonists, the presence of blood or reading the test earlier than recommended
Culture	- Highest specific method	- Not very accessible - Expensive - Transporting samples and culture of *H. pylori* are highly complex
Molecular methods	- Could be applied to multiple samples, such as gastric biopsy specimens, saliva, stool or gastric juice - Provides the antibiotic sensitivity of *H. pylori* to some antibiotics - No need for special processing supplies or transportation	- Not very accessible - Expensive - Currently only gastric samples provide good sensitivity and specificity - Only analyzes a few specific mutations for resistance to clarithromycin, tetracycline and levofloxacin - A new technology and currently still in development

EIA: enzyme immunoassay; ICA: rapid immunochromatography assay; PCPs: primary care physicians; PPI: proton pump inhibitors.

**Table 2 diagnostics-13-01918-t002:** Indications of eradication according to the III Spanish Consensus Conference on *Helicobacter pylori* (*H. pylori*) infection, Ref. [31].

Non-investigated dyspepsia in patients under 55 years old patients without alarm symptoms (+)
Functional dyspepsia
Peptic ulcer
Personal history of peptic ulcer and treatment with NSAID aspirin (long-term treatment)
Gastric cancer
Family history of (first-degree) gastric cancer
Gastric atrophy or intestinal metaplasia
Gastric (low-grade) MALT lymphoma
Iron deficiency anemia of uncertain aetiology anemia
Idiopathic thrombocytopenic purpura
Vitamin B deficiency of uncertain aetiology

Eradiation treatment must be offered to all patients with confirmed *H. pylori* infection. MALT: mucosa-associated lymphoid tissue; NSAID: non-steroidal anti-inflammatory drugs. +: alarm symptoms (anemia, bleeding, dysphagia, persistent vomiting, abdominal mass loss, and weight loss). Adapted from Ref. [31].

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
