# Peer review of "Diagnosis and Treatment of Helicobacter pylori Infection in Real Practice—New Role of Primary Care Services in Antibiotic Resistance Era"

_diagnostics, 2023, doi:10.3390/diagnostics13111918_

Round 1

Reviewer 1 Report

- H. pylori should be stated with italic font through the text.

- the purpose of study should e better highlighted in last sentences of introduction.

- In "3.3. Molecular methods" the author should also discuss about new molecular test wich detect H. pylori infection as well as its antibiotic resistance such as PCR-based method that diagnosed H. pylori and its clarhytromycin resistance from fecal clinical samples.

- The author need discuss about H. pylori heteroresistance; for example

a) Keikha M, Karbalaei M. Prevalence of antibiotic heteroresistance associated with Helicobacter pylori infection: A systematic review and meta-analysis. Microbial Pathogenesis. 2022 Aug 12:105720.

- Discuss about disadvantage of each diagnostic method using a comprehensive table.

- Conclusion should be objective with further perspectives.

Author Response

Dear Editorial Office of Diagnostics,

We are grateful for this comprehensive review of our manuscript. We have read with interest the comments and suggestions from the reviewers and the editor.

We have revised our manuscript accordingly, and we hope you will find it suitable for publication in Diagnostics.

Here we submit a point by point reply to the comments from the reviewers, along with a new version of our manuscript.

Reviewer: 1
Comments to the Author

  1. H. pylori should be stated with italic font through the text.

We have revised the article and corrected the errors

  1. The purpose of study should be better highlighted in last sentences of introduction.

As the reviewer suggests, we have added at the end of the introduction the aim of the study, which consisted of analyzing the different aspects of Helicobacter pylori infection management in real practice. We analysed the different diagnostic methods, the problem of antibiotic resistance and infection management by Gastroenterologists and mainly Primary Care physicians. 

  1. In "3. Molecular methods" the author should also discuss about new molecular test which detect H. pylori infection as well as its antibiotic resistance such as PCR-based method that diagnosed H. pylori and its clarhytromycin resistance from fecal clinical samples.

We would like to thank the reviewer for the efforts made towards improving our manuscript. A new paragraph and new references have been added in order to explain this promising diagnostic test based on faecal samples.

  1. The author need discuss about H. pylori heteroresistance; for example a) Keikha M, Karbalaei M. Prevalence of antibiotic heteroresistance associated with Helicobacter pylori infection: A systematic review and meta-analysis. Microbial Pathogenesis. 2022 Aug 12:105720.

We thank the reviewer for this suggestion and provided bibliography. We have added a paragraph explaining in detail hetero-resistance and its important role in antibiotic resistance development in H. pylori infection.

  1. Discuss about disadvantage of each diagnostic method using a comprehensive table.

We have added a table that specifies the advantages and disadvantages of each currently available diagnostic method for the diagnosis of H. pylori

  1. Conclusion should be objective with further perspectives.

We thank the suggestion. However, we consider that conclusion already mention these ideas suggested by the reviewer. It includes the current problem of antibiotic resistances and the suboptimal management of infection in daily clinical practice. Future perspectives are also described in order to improve this situation, such as the need to implement and develop better molecular diagnostic methods and to develop new strategies to improve infection management.

We would like to thank the reviewer for this thorough evaluation and the efforts made towards improving our manuscript. We largely agree with the suggestions made and considered them in a revised version of the manuscript.

Reviewer 2 Report

This is a review about diagnosis and treatment of H. pylori infection and also about antibiotic resistance.
You have not specified anywhere what type of study it is. Including in the title this detail can be added.
You also did not clearly state in the article what the purpose of this study is.
H. pylori is not written in italics everywhere. In some places it does not appear abbreviated. Please standardize its appearance in the text.
In figure 1, in the second blue shape... correct "clarithroycin".
In "Strategies to improve H. pylori infection diagnosis and treatment" you should propose/recommend strategies applicable to the Primary Care Service (this îs your title), not discuss possible limitations and problem în the hospitals.
The references are appropriate, the article presents 49 references, being up to date. Reference 29 is a self-citation, correctly used in the article. However, for a review, I recommend using more references.

Author Response

Dear Editorial Office of Diagnostics,

We are grateful for this comprehensive review of our manuscript. We have read with interest the comments and suggestions from the reviewers and the editor.

We have revised our manuscript accordingly, and we hope you will find it suitable for publication in Diagnostics.

Here we submit a point by point reply to the comments from the reviewers, along with a new version of our manuscript.

Reviewer: 2
Comments to the Author

This is a review about diagnosis and treatment of H. pylori infection and also about antibiotic resistance.

  1. You have not specified anywhere what type of study it is. Including in the title this detail can be added. You also did not clearly state in the article what the purpose of this study is.

As the reviewer suggests, we have added at the end of the introduction the type of study and the aim of this review, which consisted of analyzing the different aspects of the Helicobacter pylori infection management in real practice. We analysed the different diagnostic methods, the problem of antibiotic resistance and diagnosis and treatment of H. pylori infection by Gastroenterologists and mainly Primary Care physicians. 

  1. H. pylori is not written in italics everywhere. In some places it does not appear abbreviated. Please standardize its appearance in the text.

We have revised the article, standardize its appearance and corrected the errors

  1. In figure 1, in the second blue shape... correct "clarithroycin".

We have corrected this mistake

  1. In "Strategies to improve  pyloriinfection diagnosis and treatment" you should propose/recommend strategies applicable to the Primary Care Service (this is your title), not discuss possible limitations and problem in the hospitals.

We thank the reviewer for this suggestion. The aim of the review is to describe current scientific evidence on the new role of Primary Care physicians in H.pylori infection but also the problem of antibiotic resistance. Infection management is becoming complex due to the increase in antibiotic resistance and in new consensus documents eradication regimens that are also prescribed at Primary Care Level have had to adapt to this reality. Therefore, it is necessary to investigate local antibiotic resistance and adapt antibiotic treatments to those data.  So far, we only had available the culture, a technique with multiple limitations.  Nowadays, molecular methods are positioning as a very useful tool to detect individual antibiotic resistance and prescribe targeted antibiotic treatments but they still need to improve its sensibility and specificity in non-invasive samples (such as fecal samples) to be implemented at Primary Care Level. For this reason, we believe that it could be interesting to mention in this section strategies to improve the diagnostic of antibiotic resistance as a first strategy to improve H. pylori infection diagnosis and treatment.

We have introduced some modifications at this point to clarify it.

  1. The references are appropriate, the article presents 49 references, being up to date. Reference 29 is a self-citation, correctly used in the article. However, for a review, I recommend using more

As the reviewer suggests, some new references have been added.

We would like to thank the reviewer for this thorough evaluation and the efforts made towards improving our manuscript. We largely agree with the suggestions made and considered them in a revised version of the manuscript.